# Definer: A computational method for accurate identification of RNA pseudouridine sites based on deep learning

Bo Han[1], Sudan Bai[2], Yang Liu[1], Jiezhang Wu[2], Xin Feng🄯[3]*, Ruihao Xin[2]*

**1** Jilin Chemical Hospital, Jilin, P.R. China, **2** College of Information and Control Engineering, Jilin Institute of Chemical Technology, Jilin, P.R. China, **3** School of Science, Jilin Institute of Chemical Technology, Jilin, P.R. China

* fengxin@jlict.edu.cn (XF); xinruihao@jlict.edu.cn (RX)

## Abstract

Pseudouridine is an important modification site, which is widely present in a variety of non-coding RNAs and is involved in a variety of important biological processes. Studies have shown that pseudouridine is important in many biological functions such as gene expression, RNA structural stability, and various diseases. Therefore, accurate identification of pseudouridine sites can effectively explain the functional mechanism of this modification site. Due to the rapid increase of genomics data, traditional biological experimental methods to identify RNA modification sites can no longer meet the practical needs, and it is necessary to accurately identify pseudouridine sites from high-throughput RNA sequence data by computational methods. In this study, we propose a deep learning-based computational method, Definer, to accurately identify RNA pseudouridine loci in three species, *Homo sapiens, Saccharomyces cerevisiae* and *Mus musculus*. The method incorporates two sequence coding schemes, including NCP and One-hot, and then feeds the extracted RNA sequence features into a deep learning model constructed from CNN, GRU and Attention. The benchmark dataset contains data from three species, *H. sapiens*, *S. cerevisiae* and *M. musculus*, and the results using 10-fold cross-validation show that Definer significantly outperforms other existing methods. Meanwhile, the data sets of two species, *H. sapiens* and *S. cerevisiae*, were tested independently to further demonstrate the predictive ability of the model. In summary, our method, Definer, can accurately identify pseudouridine modification sites in RNA.

## 1. Introduction

RNA modification is an important component of gene regulation and is involved in various biological processes [1,2]. To date, over 150 types of RNA modifications have been discovered in the field of biology [3,4]. Among them, pseudouridine (Ψ) modification is the earliest and most abundant RNA modification found in various types of RNA, including mRNA, tRNAs, and snRNA, etc [5,6]. The most common processes of RNA modification are pseudouridylation and methylation [7]. Studies have shown that pseudouridine can change the secondary and tertiary structure of RNA, affect the speed of gene expression, and is closely related to various diseases, such as Parkinson's disease, congenital keratinization disorder, and myelodysplastic syndrome keratosis, which are associated with pseudouridine modification mutations [8,9].

**Data availability statement:** The datasets for the three species, the Definer-PseU model code, and the Software engineering code are available for free download on the https://github.com/louise1223/Definer.

**Funding:** This work is supported by the Natural Science Foundation of Jilin Province (YDZJ202301ZYTS401, YDZJ202301ZYTS288), the National Natural Science Foundation of China, Mathematical Tianyuan Fund Project (12326377). The funders had no role in study design, data collection and analysis, decision to publish, or preparation of the manuscript.

**Competing interests:** The authors have declared that no competing interests exist.

Therefore, studying pseudouridine modification sites is of great significance for both biology and medicine [10,11]. With the advent of the post-genomic era, the amount of genomic data has increased rapidly, and traditional biological experiments are no longer able to meet the actual research needs [12]. Therefore, it is necessary to develop more convenient computational models to extract information on pseudouridine sites and accurately identify them [13,14].

Many computational methods based on machine learning and deep learning have been developed for predicting pseudouridine sites in three species, including *H. sapiens*, *S. cerevisiae*, and *M. musculus*. Li et al. [15] constructed the first prediction model for pseudouridine sites, PPUS, based on the SVM algorithm for predicting Homo sapiens and Saccharomyces cerevisiae through a web server. Chen et al. [16] constructed datasets for *H. sapiens*, *S. cerevisiae*, and *Mus musculus* and combined the PseDNC encoding method with SVM to build the iRNA-PseU prediction model. The constructed dataset was further used in subsequent research. He et al. [17] proposed PseUI, which combines five encoding methods with SVM algorithm and further improves the accuracy of pseudouridine site recognition by applying sequence forward feature selection. Subsequently, Tahir et al. [18] used the One-hot encoding method and built a two-layer convolutional neural network to develop iPseU-CNN. Liu et al. [19] developed XG-PseU, a prediction method based on extreme Gradient Boosting (XGBoost). Bi et al. [20] discovered the ensemble learning algorithm, which integrates five different machine learning classifiers to build the EnsemPseU predictor for predicting pseudouridine sites. Lv et al. [21] used the random forest algorithm combined with the light gradient boosting machine algorithm and the incremental feature selection strategy to build a new predictor, RF-PseU, which improved prediction performance. Mu et al. [22] constructed MU-PseUDeep by combining the original sequence and secondary structure with a convolutional neural network, further improving the performance of predicting pseudouridine sites. Song et al. [23,24] constructed PIANO and PSI-MOUSE predictors based on genomic and sequence features for predicting pseudouridine sites. This is the first time that genomic-derived features have been introduced and achieved good performance in predicting pseudouridine sites. Li et al. [25] developed the porpoise predictor based on the stacked ensemble learning method and used four feature selection methods. Table 1 summarizes the existing pseudouridine site predictors, including benchmark datasets, feature extraction, classifiers, performance evaluation, and network servers, and most of these computational methods predict the three species, *H. sapiens*, *S. cerevisiae*, and *M. musculus*, with only PPUS predicting *H. sapiens* and *S. cerevisiae*.

Although previous studies have made significant contributions and provided a foundation for subsequent research, there is still considerable room for improvement in predicting RNA sequence performance based on existing methods [26,27]. Developing better prediction methods will enable a comprehensive understanding of the relationship between RNA sequences and life activities. Existing prediction tools mostly rely on a single feature extraction algorithm and traditional machine learning algorithms [28,29]. Due to the extremely complex sequence features exhibited by biological sequences, traditional machine learning methods cannot achieve better prediction performance. Deep learning algorithms have strong learning and generalization abilities, and possess good modeling capabilities [30]. Therefore, it is necessary to improve prediction performance by increasing sequence features and developing more suitable classification algorithms [31].

Based on the above issues, this paper proposes a deep learning-based computational method, Definer, to identify pseudouridine (Ψ) sites in three species, including *H. sapiens*, *S. cerevisiae*, and *M. musculus*. Firstly, we combined the One-hot and NCP feature encoding schemes to extract RNA sequence information. Secondly, we constructed Ψ site prediction models based on three deep learning models: convolutional neural network (CNN), gated recurrent unit (GRU), and attention mechanism. Finally, ten-fold cross-validation and independent testing showed that, compared with state-of-the-art methods, Definer significantly improved the prediction performance on Ψ site identification in all three species.

**Table 1. Summary of existing methods for RNA pseudouridine site prediction.**

| Tools[a] | Species | Encoding | Classifier | Evaluation strategy | Webserver/software[b] |
|---|---|---|---|---|---|
| PPUS [15] | *H. sapiens* *S. cerevisiae* | Binary | SVM | 5-fold CV | Yes |
| iRNA-PseU [16] | *H. sapiens* *S. cerevisiae* *M. musculus* | PseKNC | SVM | jackknife test and independent test | Yes |
| PseUI [17] | *H. sapiens* *S. cerevisiae* *M. musculus* | NAC,DNC, PseDNC, PSNP, PSDP | SVM | jackknife test and independent test | Yes |
| iPseU-CNN [18] | *H. sapiens* *S. cerevisiae* *M. musculus* | One-hot | CNN | 5-fold CV | No |
| XG-PseU [19] | *H. sapiens* *S. cerevisiae* *M. musculus* | NAC,DNC,TNC, NCP,ND, One-hot | XGBoost | 10-fold CV and independent test | Yse |
| EnsemPseU [20] | *H. sapiens* *S. cerevisiae* *M. musculus* | Kmer,Binary, ENAC,NCP, ND | Ensemble | 10-fold CV and independent test | No |
| RF-PseU [21] | *H. sapiens* *S. cerevisiae* *M. musculus* | Binary,ANF, NCP,EIIP,ENAC, CKSNAP | RF | 10-fold CV and independent test | Yes |
| MU-PseUDeep [22] | *H. sapiens* *S. cerevisiae* *M. musculus* | One-hot,PSNP | CNN | 10-fold CV and independent test | No |
| PIANO [23] | *H. sapiens* *S. cerevisiae* *M. musculus* | SCP,PSNP, Genome-derived features | SVM | 10-fold CV and independent test | Yes |
| PSI-MOUSE [24] | *H. sapiens* *S. cerevisiae* *M. musculus* | NCP,ND, Genome-derived features | SVM | 10-fold CV and independent test | Yes |
| Porpoise [25] | *H. sapiens* *S. cerevisiae* *M. musculus* | Binary,PseKNC, NCP,PSTNPss | SVM,GBDT, GaussianNB, XGBoost | 10-fold CV and independent test | Yes |

Abbreviations: Binary, binary features; PseKNC, pseudo nucleotide composition; NAC, nucleic acid composition; DNC,di-nucleotide composition; PseDNC, pseudo nucleic acid composition; PSNP, position-specific nucleotide propensity; PSDP, position-specific dinucleotide propensity; TNC, tri-nucleotide composition; ENAC, enhanced nucleic acid composition; ANF, accumulated nucleotide frequency; EIIP, electron-ion interaction pseudopotentials of trinucleotide; CKSNAP, composition of *k*-spaced nucleic acid pairs; SCP, structural chemical properties; SVM, support vector machine; CNN, convolutional neural networks; XGBoost, extreme Gradient Boosting; RF, random forest; GBDT, gradient boosting decision tree; GaussianNB, gaussian naive bayes; CV, cross-validation.

[a]The URL addresses for the listed tools are as follows: PPUS, http://lyh.pkmu.cn/ppus/; iRNA-PseU, http://lin.uestc.edu.cn/server/iRNA-PseU; PseUI, http://zhulab.ahu.edu.cn/PseUI/; XG-PseU, http://www.bioml.cn/; RF-PseU, http://rfpsu.aibiochem.net/; PIANO, http://piano.rnamd.com/; PSI-MOUSE, http://piano.rnamd.com/.

[b]Yes: the publication is accompanied with a webserver/soft package and it is still functional; No: the publication has no webserver or soft package.

## 2. Materials and methods

### 2.1. Overall framework

The experimental design process and performance evaluation of this study are shown in Fig 1, which includes five main steps: data collection, feature extraction, model construction, performance evaluation, and visualization software development [32]. Firstly, benchmark datasets and independent test sets for three species, H. sapiens, S. cerevisiae, and M. musculus, were collected from relevant literature and public databases [24]. Two feature extraction methods were then employed to extract sequence information from the datasets. Subsequently, a deep learning-based predictor, Definer, was constructed, which achieved good performance on all three species. Furthermore, we evaluated and compared

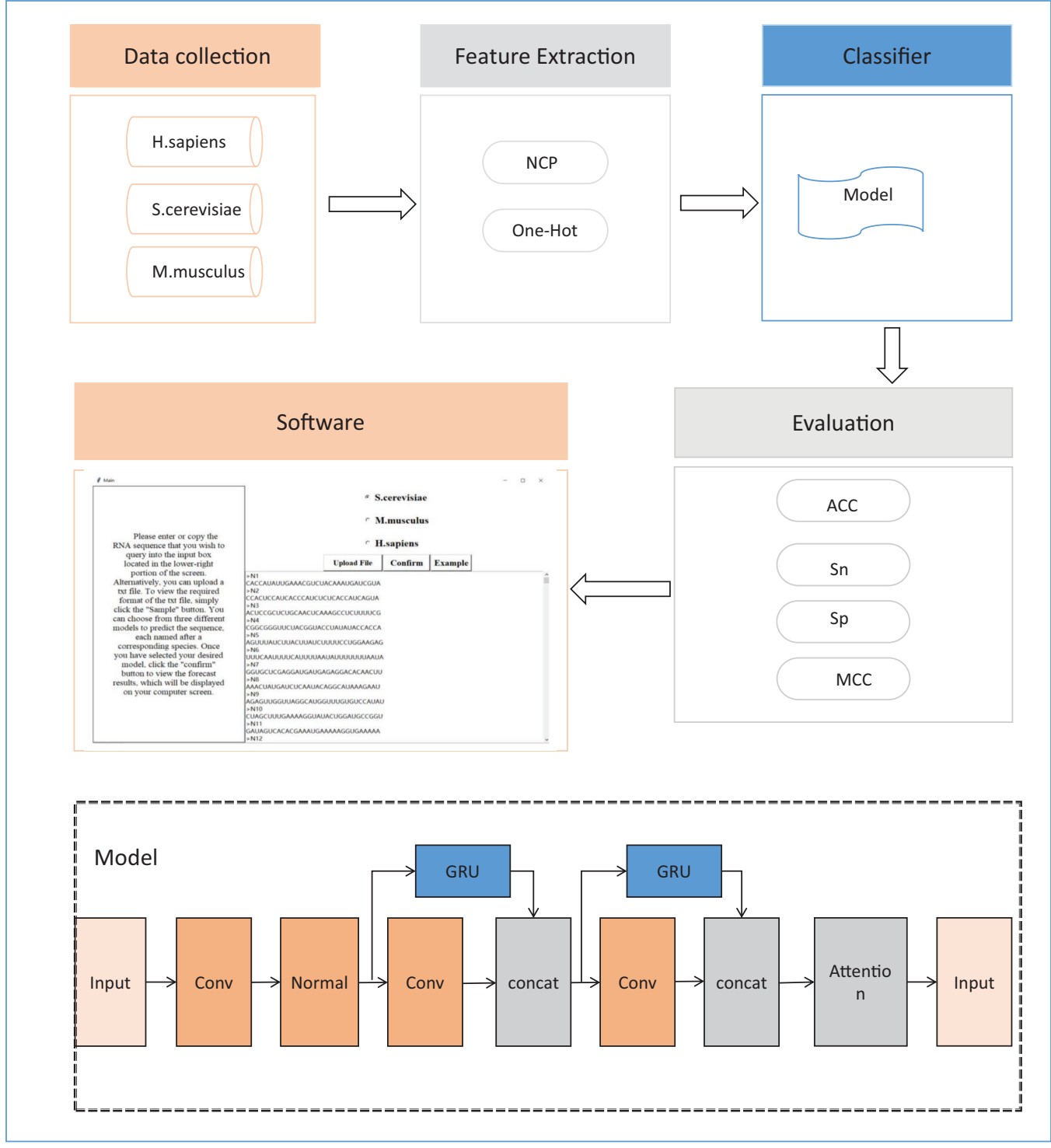

**Fig 1. Shows the experimental design process.**

our Definer with several existing methods, and found that its prediction performance was significantly improved. Finally, we developed and made publicly available a software for users to utilize online.

## 2.2. Benchmark data sets

In order to facilitate comparison with existing methods, we used the dataset constructed by Chen et al. (11), which is commonly used in most of the existing prediction methods, such as iRNA-PseU [15], PseUI [16], iPseU-CNN [17], XG-PseU [18], and Porpoise [25]. The datasets for the three species were obtained from the RMBase database [33], including three benchmark datasets *H. sapiens* (H_990), *S. cerevisiae* (S_628), and *M. musculus* (M_994), which were used for model training, and two independent test sets, which only included *H. sapiens* (H_200) and *S. cerevisiae* (S_200) species. The details of the datasets are shown in Table 2.

## 2.3. Feature extraction

Feature extraction is an important step in building a prediction model, which aims to encode RNA sequence fragments containing only four nucleotides, adenine (A), cytosine (C), guanine (G), and uracil (U), into digitized feature vectors [34]. The way of extracting input data has a great impact on the model. Only by choosing a suitable feature extraction method according to specific conditions can better training results be achieved. Efficient feature extraction methods can effectively extract more representative feature vectors and provide strong support for subsequent model construction [35]. In this study, we used two feature extraction methods, including One-hot encoding and nucleotide chemical properties (NCP). Brief introductions of these two feature extraction methods are presented below.

**2.3.1. One-hot encoding.** One-hot encoding is a binary encoding method and one of the basic feature representation methods for RNA sequences. The basic idea of one-hot encoding is to convert each base in the sequence into a four-dimensional binary vector, where only one dimension is 1 and the others are 0. The four nucleotides A, U, C, and G will be respectively converted into vectors (1,0,0,0), (0,1,0,0), (0,0,1,0), and (0,0,0,1) [36].

**2.3.2. NCP.** The RNA sequence is composed of four nucleotides: adenine (A), cytosine (C), guanine (G), and uracil (U). These nucleotides have different structures and chemical properties. Nucleotide Chemical Property (NCP) encodes RNA sequences by three different chemical properties, including cyclic structure, hydrogen bonding, and chemical functionality [37]. Regarding cyclic structure, A and G are purines with two rings, while C and U are pyrimidines with one ring. Concerning hydrogen bonding, A and U form two hydrogen bonds during hybridization, while G and C can form three hydrogen bonds [38]. Regarding chemical functionality, A and C contain an amino group, while G and U contain a ketone base. Based on the three different chemical structural properties, nucleotides in RNA sequences can be represented by a vector $L_j = \left( X_j, Y_j, Z_j \right)$, where X represents cyclic structure, Y represents hydrogen bonding, and Z represents chemical functionality. The feature representation method of NCP is shown in equation (1).

**Table 2. Datasets introduction.**

| Datasets | Number of positive samples | Number of negative samples | Sequence length |
|---|---|---|---|
| H_990 | 495 | 495 | 21 |
| S_628 | 314 | 314 | 31 |
| M_994 | 497 | 497 | 21 |
| H_220 | 100 | 100 | 21 |
| S_220 | 100 | 100 | 31 |

$$\begin{cases} X_j = \begin{cases} 1, \text{ if } L_j \in (A,G) \\ 0, \text{ if } L_j \in (C,U) \end{cases} \\ Y_j = \begin{cases} 1, \text{ if } L_j \in (A,C) \\ 0, \text{ if } L_j \in (G,U) \end{cases} \\ Z_j = \begin{cases} 1, \text{ if } L_j \in (A,U) \\ 0, \text{ if } L_j \in (C,G) \end{cases} \end{cases} \tag{1}$$

## 2.4. Deep learning model framework

This study aims to construct a prediction model for pseudouridine sites in RNA based on three classical deep learning models: convolutional neural network (CNN), gated recurrent unit (GRU), and attention. Firstly, the input data is processed through the first convolution layer of the CNN, which performs cross-correlation operations on the matrix of each channel from left to right and top to bottom using convolution kernels. Then, the obtained data is regularized to prevent overfitting, as shown in equations (2) and (3).

$$Conv(\mathrm{x}) = \sum_{p=1}^{P}\sum_{q=1}^{Q} w_{pq}^k x_i + p,q \tag{2}$$

$$L = E_{in} + \lambda \sum_j \left| W_j \right| \tag{3}$$

Next, the data and parameters are compressed through the pooling layer of CNN, and the compressed data is fed into the second layer of CNN. At the same time, the feature tensor is fused with GRU and the Relu activation function is used to accelerate the convergence of the model, as shown in equation (4). The update gate $Z_t$ is used to filter information, and $W_z$ controls the retention level of new and old information input at each time step. The reset gate $r_t$ is used to filter information, and $W_r$ controls the retention level of input information at each position at time t-1 [39].

$$\begin{cases} Z_t = \sigma\left(W_z\left[h_{t-1}, x_t\right]\right) \\ r_t = \sigma\left(W_r\left[h_{t-1}, x_t\right]\right) \\ \tilde{h}_t = \tan h\left(W\left[r_t * h_{t-1}, x_t\right]\right) \\ h_t = (1 - Z_t) * h_{t-1} + z_t * \tilde{h}_t \end{cases} \tag{4}$$

Then, following the same method of combining the second layer convolution of CNN with GRU, the feature tensors of the third layer convolution of CNN and GRU are fused. Finally, to focus on important information and fully absorb it, an Attention mechanism is added to the model.

## 2.5. Performance evaluations

Model evaluation is an important step to verify the feasibility of a model, and there are three commonly used methods: K-fold cross-validation, independent testing, and overlapping checking. In order to facilitate comparison with existing methods and better demonstrate the effectiveness of the proposed method, we choose to use the first two methods for evaluation, respectively based on the training dataset and the testing dataset using 10-fold cross-validation and independent tests. For a classification task, accuracy (ACC) is the most basic evaluation

index, which represents the percentage of correct classification. However, the basic evaluation index often cannot reflect the model's performance well, which may lead to poor judgments. This study uses four evaluation indicators, including specificity (Sp), sensitivity (Sn), accuracy (ACC), and Matthew's correlation coefficient (MCC) to evaluate the predictive model [40,41]. The calculation formulas for the four evaluation indicators are shown below.

$$S_n = \frac{TP}{TP + FN} \tag{5}$$

$$S_p = \frac{TN}{TN + FP} \tag{6}$$

$$ACC = \frac{TP + TN}{TP + TN + FP + FN} \tag{7}$$

$$MCC = \frac{TP \times TN - FP \times FN}{\sqrt{(TP + FP)(TP + FN)(TN + FP)(TN + FN)}} \tag{8}$$

In this paragraph, TP, FP, TN, and FN respectively represent the number of true positives, false positives, true negatives, and false negatives.

## 3. Results and discussion

### 3.1. Distribution of nucleotide positions at pseudouridine sites

To analyze the characteristics of pseudouridine sites in RNA sequences, we used Two Sample Logo [42] to calculate the importance of nucleotides at each position. Two Sample Logo is a tool for computing differences between nucleotide samples and the significance of nucleotides at each position in a sequence. The nucleotide distributions of pseudouridine sites in the *H. sapiens*, *S. cerevisiae*, and *M. musculus* species are shown in Fig 2a, 2b, and 2c, respectively. The size of each letter represents the frequency of the corresponding base at that position, with larger letters indicating higher frequencies. At each position, the letters are arranged in order of dominance from top to bottom, with the most dominant base at the top. From the figures, it can be seen that in *H. sapiens*, uridine (U) content is highest near the central position 11, while cytidine (C) is mainly distributed at downstream positions 17 and 20. In *S. cerevisiae*, guanine (G) is mostly distributed in the upper-middle region, while uridine (U) is distributed at the central positions 14, 15, and 16. Adenine (A) is mainly distributed in the upper-middle region, with three consecutive A bases at positions 13, 14, and 15. In *M. musculus*, uridine (U) is distributed in the upper-middle region, with U bases at positions 9, 11, 12, and 13. These results indicate that there are different nucleotide distribution patterns between pseudouridine and non-pseudouridine sites in the *H. sapiens*, *S. cerevisiae*, and *M. musculus* species, and therefore, it is necessary to establish a universal prediction model across different species.

### 3.2. Performance comparison analysis of different feature extraction methods

An efficient feature extraction method can effectively extract more representative feature vectors and provide strong support for subsequent model construction. This section compared One-hot, NCP, and their fusion, respectively, by placing these three feature extraction methods into the predictor for comparison. The comparative results for the three species *H.*

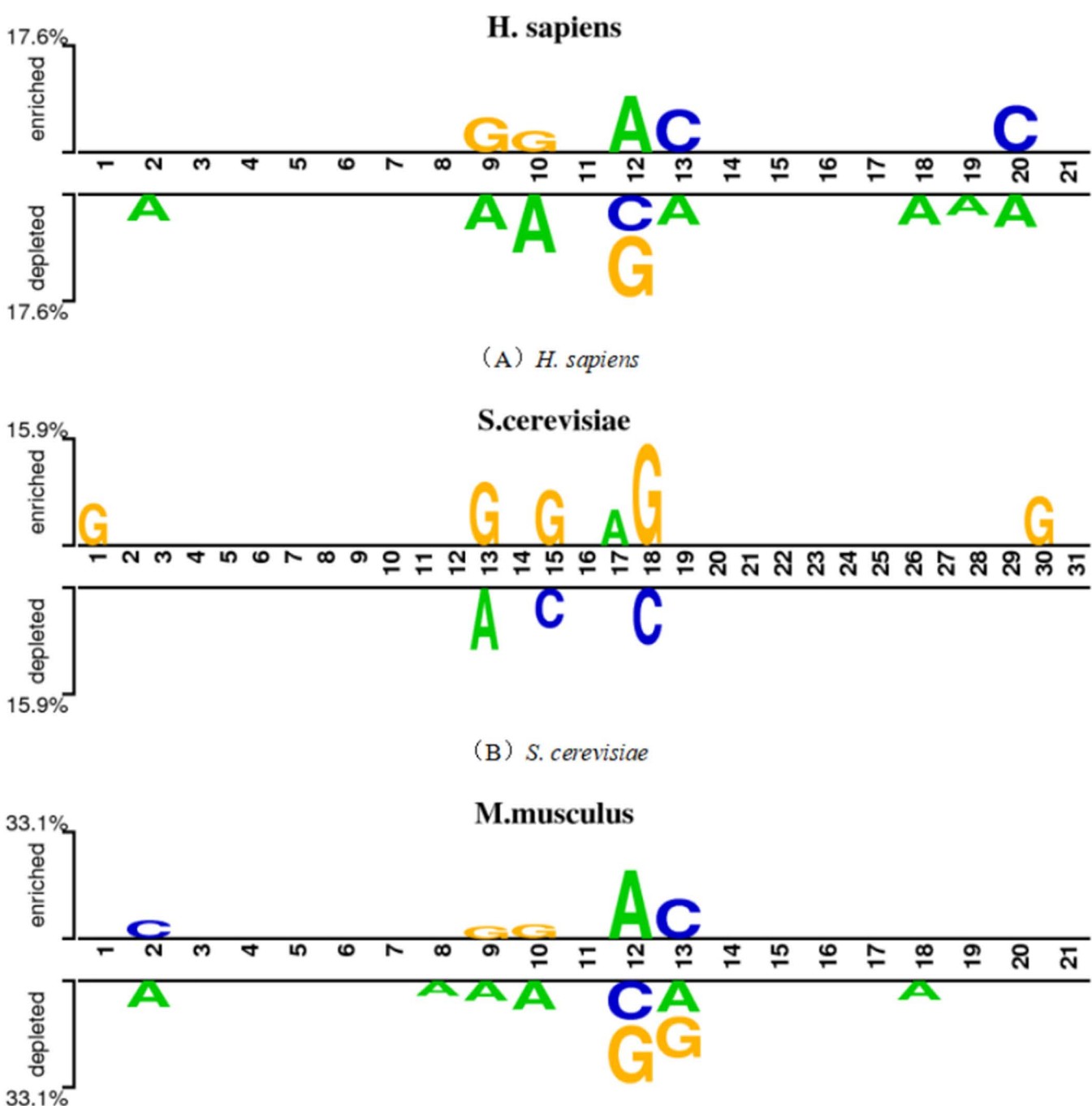

**Fig 2. Distribution of nucleotide positions of the three species.**

sapiens, *S. cerevisiae*, and *M. musculus* are shown in Tables 3–5, respectively. Please note that there are no grammar errors in the original text.

From the table, it becomes apparent that the combination of both encoding methods yields superior results across all three datasets in comparison to the utilization of a single feature extraction approach. In the context of the H_990 dataset, the amalgamation of One-hot and NCP surpasses both individual feature extraction methods with respect to all four evaluation

**Table 3. Three feature extraction methods used in H_990 comparison of results.**

| Feature extraction | ACC (%) | Sn (%) | Sp (%) | MCC (%) |
|---|---|---|---|---|
| One-hot | 81.02 | 83.25 | 78.78 | 62.38 |
| NCP | 76.55 | 76.77 | 76.33 | 53.48 |
| One-hot+NCP | **82.95** | **85.28** | **80.62** | **66.43** |

Note: Bold numbers indicate the maximum value in this column.

**Table 4. Three feature extraction methods used in S_628 comparison of results.**

| Feature extraction | ACC (%) | Sn (%) | Sp (%) | MCC (%) |
|---|---|---|---|---|
| One-hot | 85.81 | 85.00 | 86.62 | 72.18 |
| NCP | 84.52 | **86.28** | 82.77 | 69.91 |
| One-hot+NCP | **86.01** | 84.05 | **87.97** | **72.66** |

Note: Bold numbers indicate the maximum value in this column.

**Table 5. Three feature extraction methods used in M_944 comparison of results.**

| Feature extraction | ACC (%) | Sn (%) | Sp (%) | MCC (%) |
|---|---|---|---|---|
| One-hot | 86.93 | **88.32** | 85.54 | 74.03 |
| NCP | 85.76 | 85.97 | 85.56 | 71.76 |
| One-hot+NCP | **87.15** | 87.04 | **87.26** | **74.41** |

Note: Bold numbers indicate the maximum value in this column.

metrics. Specifically, it attains an accuracy rate of 82.95%, which represents a notable enhancement of 1.93% over the accuracy achieved by the One-hot method alone. While it is true that the single feature extraction method Sn exhibits a relatively better performance than the fusion on the S_628 and M_944 datasets in terms of a particular aspect, it is important to note that the fusion method demonstrates a significantly more favorable performance in the other three evaluation metrics. The fused ACC values on the S_628 and M_944 datasets are 86.01% and 87.15%, respectively. Based on these comprehensive observations and analyses, we have opted to employ the fusion of One-hot and NCP for the extraction of sequence feature information pertaining to pseudouridine sites, as it offers a more comprehensive and effective means of capturing the essential characteristics and patterns within the data.

## 3.3. Performance comparison analysis of different models

The classifier is an important component of the experiment and is closely related to the final experimental results. Building a suitable predictive model can greatly improve experimental performance. Deep learning is a machine learning algorithm with feature learning ability. It can extract and learn low-level data features to obtain more abstract high-level features. In recent years, the genomics databases have grown rapidly. Only by using classifiers with stronger learning ability can we better learn and mine effective information in huge databases [43]. In this section, a predictor called Definer was constructed based on commonly used deep learning algorithms. We compared it with several traditional machine learning algorithms and commonly used deep learning algorithms, including SVM, RF, LightGBM, and CNN, based on ten-fold cross-validation on three benchmark datasets. The comparison results of the three benchmark datasets are shown in Fig 3. From the figure, it can be seen that the evaluation

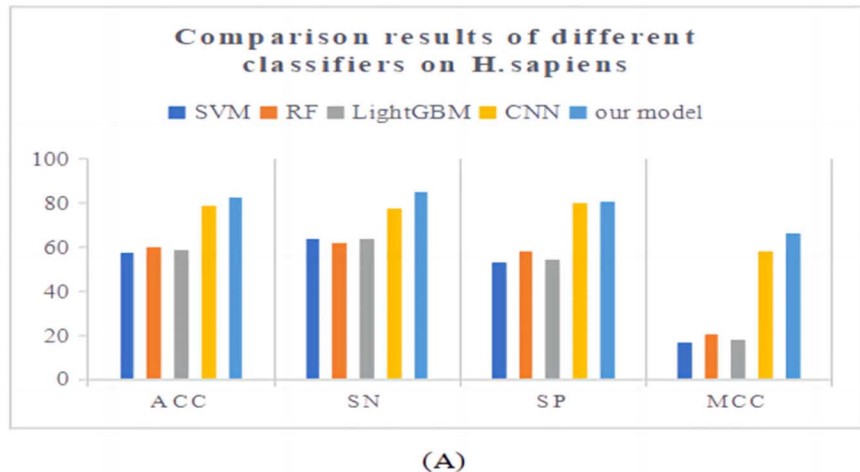

(A)

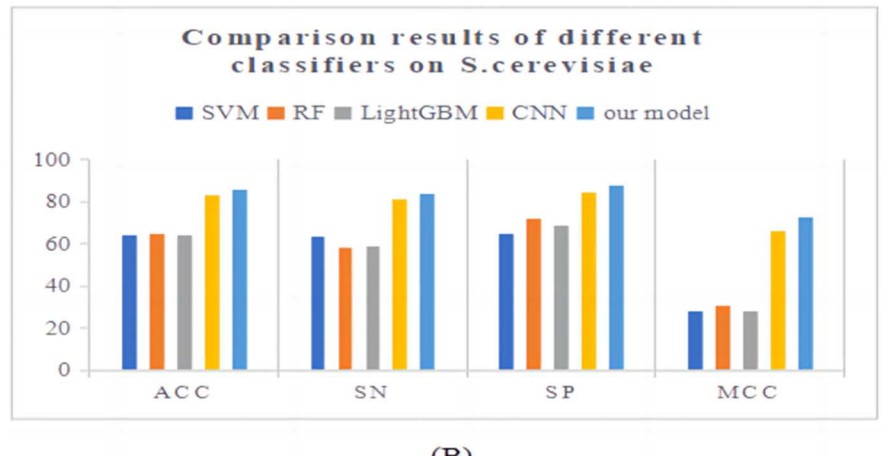

(B)

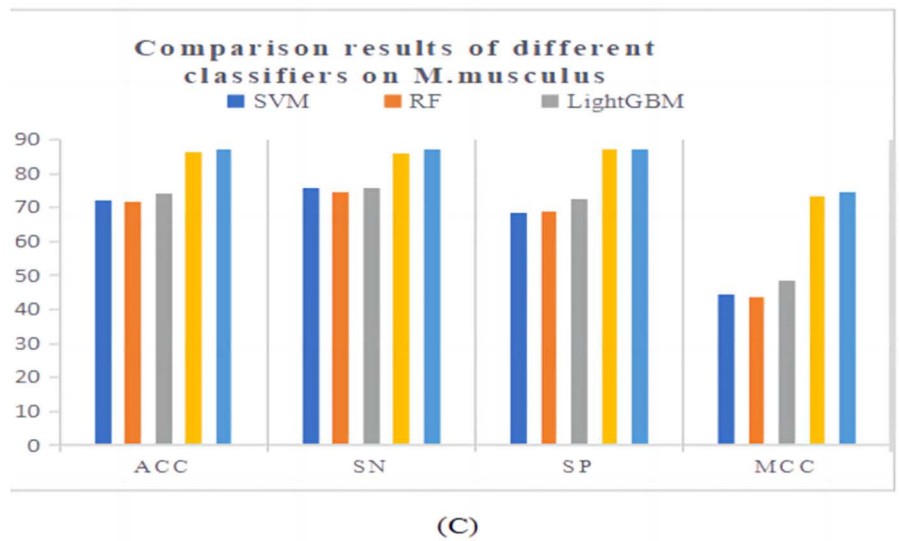

(C)

**Fig 3. The comparison results of the three benchmark dataset.**

indicators of deep learning algorithms are much higher than those of traditional machine learning algorithms. We built a new predictor Definer based on CNN, which integrates two GRUs and introduces Attention. From the prediction results, our model is superior to other classifiers in all four evaluation indicators.

## 3.4. Comparative analysis on independent datasets

To further substantiate the proficiency of Definer in predicting pseudouridine sites, this section undertakes a comprehensive verification and evaluation process using two independent test sets, which encompass two distinct species, namely Homo sapiens and Saccharomyces cerevisiae. Concurrently, an in-depth comparison of the predictive performance of our proposed method with that of several pre-existing methods was carried out on these two independent test sets, and the corresponding results are meticulously presented in Table 6. As is conspicuously demonstrated in the table, our predictor exhibits a highly significant and preponderant performance, outstripping other methods across all four evaluation metrics on the two independent test datasets. Particularly in the case of Homo sapiens, the achieved Accuracy (ACC) reaches an impressive 83.50%, which represents a remarkable increment of 6% over the hitherto best-performing existing method, Porpoise. This outstanding performance not only attests to the enhanced predictive power of Definer but also underlines its potential to make a substantial contribution in the realm of pseudouridine site prediction and related bioinformatics research.

## 3.5. Performance comparison with state-of-the-art methods

In this section, a comprehensive comparison was conducted between Definer and a series of state-of-the-art methods across three benchmark datasets, namely XG-PseU [19], iPseU-CNN [18], PseUI [17], iRNA-PseU [16], EnsemPseU [20], RF-PseU [21], and Porpoise [25]. The detailed comparison results are presented in Table 7. Upon performing ten-fold cross-validation on the identical two independent test datasets, it becomes evident that Definer

**Table 6. Comparison Evaluation with state-of-the-art several methods on the same independent test dataset.**

| Species | Method | ACC (%) | Sn (%) | Sp (%) | MCC (%) |
|---|---|---|---|---|---|
| *H.sapiens* | XG-PseU [19] | 67.50 | 68.00 | 67.00 | 35.00 |
| (H_200) | iPseU-CNN [18] | 69.00 | 77.72 | 60.81 | 40.00 |
| | PseUI [17] | 65.50 | 63.00 | 68.00 | 31.00 |
| | iRNA-PseU [16] | 61.50 | 58.00 | 65.00 | 23.00 |
| | EnsemPseU [20] | 69.50 | 73.00 | 66.00 | 39.00 |
| | RF-PseU [21] | 75.00 | 78.00 | 72.00 | 50.00 |
| | Porpoise [25] | 77.35 | 82.30 | 72.40 | 55.13 |
| | **Definer** | **83.50** | **91.00** | **88.00** | **73.34** |
| *S.cerevisiae* | XG-PseU [19] | 71.00 | 75.00 | 67.00 | 42.14 |
| (S_200) | iPseU-CNN [18] | 73.50 | 68.76 | 77.82 | 47.00 |
| | PseUI [17] | 68.50 | 65.00 | 72.00 | 37.00 |
| | iRNA-PseU [16] | 60.00 | 63.00 | 57.00 | 20.00 |
| | EnsemPseU [20] | 75.00 | 85.00 | 65.00 | 51.00 |
| | RF-PseU [21] | 77.00 | 75.00 | 79.00 | 54.00 |
| | Porpoise [25] | 83.50 | 88.00 | 79.00 | 67.27 |
| | **Definer** | **88.00** | **90.00** | **88.00** | **78.59** |

Note: Bold numbers indicate the maximum value in this column.

**Table 7. Comparison of three benchmark datasets with existing methods.**

| Species | Method | ACC (%) | Sn (%) | Sp (%) | MCC (%) |
|---|---|---|---|---|---|
| *H.sapiens* | XG-PseU | 65.44 | 63.64 | 67.24 | 31.00 |
| (H_990) | iPseU-CNN | 66.68 | 65.00 | 68.78 | 34.00 |
| | PseUI | 64.24 | 64.85 | 63.64 | 28.00 |
| | iRNA-PseU | 60.40 | 61.01 | 59.80 | 21.00 |
| | EnsemPseU | 66.28 | 63.46 | 69.09 | 33.00 |
| | RF-PseU | 64.30 | 66.10 | 62.60 | 29.00 |
| | Porpoise | 78.53 | **89.11** | 67.94 | 58.45 |
| | **Definer(10-fold CV)** | **85.68** | 84.66 | **86.50** | **71.31** |
| S.*cerevisiae* | XG-PseU | 68.15 | 66.84 | 69.45 | 37.00 |
| (S_628) | iPseU-CNN | 68.15 | 66.36 | 70.45 | 37.00 |
| | PseUI | 65.13 | 62.74 | 67.52 | 30.00 |
| | iRNA-PseU | 64.49 | 64.65 | 64.33 | 29.00 |
| | EnsemPseU | 74.16 | 73.88 | 74.45 | 49.00 |
| | RF-PseU | 74.80 | 77.20 | 72.40 | 49.00 |
| | Porpoise | 81.69 | 81.21 | 82.17 | 63.38 |
| | **Definer(10-fold CV)** | **86.30** | **85.68** | **86.94** | **73.01** |
| *M.musculus* | XG-PseU | 72.03 | 76.48 | 67.57 | 45.00 |
| (M_944) | iPseU-CNN | 71.81 | 74.79 | 69.11 | 44.00 |
| | PseUI | 70.44 | 74.58 | 66.31 | 41.00 |
| | iRNA-PseU | 69.07 | 73.31 | 64.83 | 38.00 |
| | EnsemPseU | 73.85 | 75.43 | 72.25 | 48.00 |
| | RF-PseU | 74.80 | 73.10 | 76.50 | 50.00 |
| | Porpoise | 77.75 | 77.83 | 77.67 | 55.55 |
| | **Definer(10-fold CV)** | **87.68** | **88.10** | **87.25** | **75.54** |

Note: Bold numbers indicate the maximum value in this column and CV indicates cross-validation.

exhibits remarkable superiority. In the case of two species, S. cerevisiae and M. musculus, Definer surpasses the other seven prediction methods with respect to all four evaluation metrics. Even in H. sapiens, although the Sn of the porpoise tool is marginally higher than that of our predictor, it is crucial to note that our predictor demonstrates a significant edge and is far more excellent in the remaining three evaluation metrics. This clearly indicates that Definer not only attains a higher level of accuracy but also showcases enhanced stability when compared to other existing methods. By integrating the comparison outcomes from the independent test set in the preceding section, it can be firmly and conclusively drawn that Definer is capable of precisely and accurately predicting pseudouridine sites within the three species, namely H. sapiens, S. cerevisiae, and M. musculus, thereby establishing its efficacy and reliability in the field of pseudouridine site prediction.

## 3.6. Software engineering

In the field of bioinformatics, there are numerous commonly used analysis methods and online tools. For example, sequence alignment tools like GGMSA [44] are widely utilized. GGMSA allows for the comparison of nucleotide or amino acid sequences, enabling the identification of homologous sequences and providing insights into evolutionary relationships and functional similarities. It uses efficient algorithms to search large sequence databases rapidly, which is a remarkable technical achievement.

Another important tool is gene expression analysis software such as DESeq2 [45]. It can analyze differential gene expression between different samples or conditions. By applying statistical models and normalization techniques, it helps to identify genes that are significantly upregulated or downregulated, which is crucial for understanding biological processes and disease mechanisms.

Recent advancements in computational methods, such as the MRNDR [46], further enhance the ability to analyze complex biological data and uncover potential drug repurposing opportunities through sophisticated attention mechanisms and deep learning architectures.

However, when it comes to the prediction of potential pseudo-uridine sites from RNA sequences, existing tools have certain limitations. While they may focus on general sequence analysis or other types of RNA modifications, they do not specifically target pseudo-uridine sites with high accuracy and user-friendly visualization.

Software engineering and web servers have become essential in the Internet age. To address the need for identifying potential pseudo-uridine sites, we have developed a software visualization based on our model Definer. It is developed using the Python Tkinter framework. The main interface of this software, as shown in Fig 4, offers a unique solution. It provides users with an intuitive and convenient way to input RNA sequences and obtain predictions of potential pseudo-uridine sites. The model Definer underlying the software has been carefully designed and trained to improve the accuracy of pseudo-uridine site prediction, filling a gap

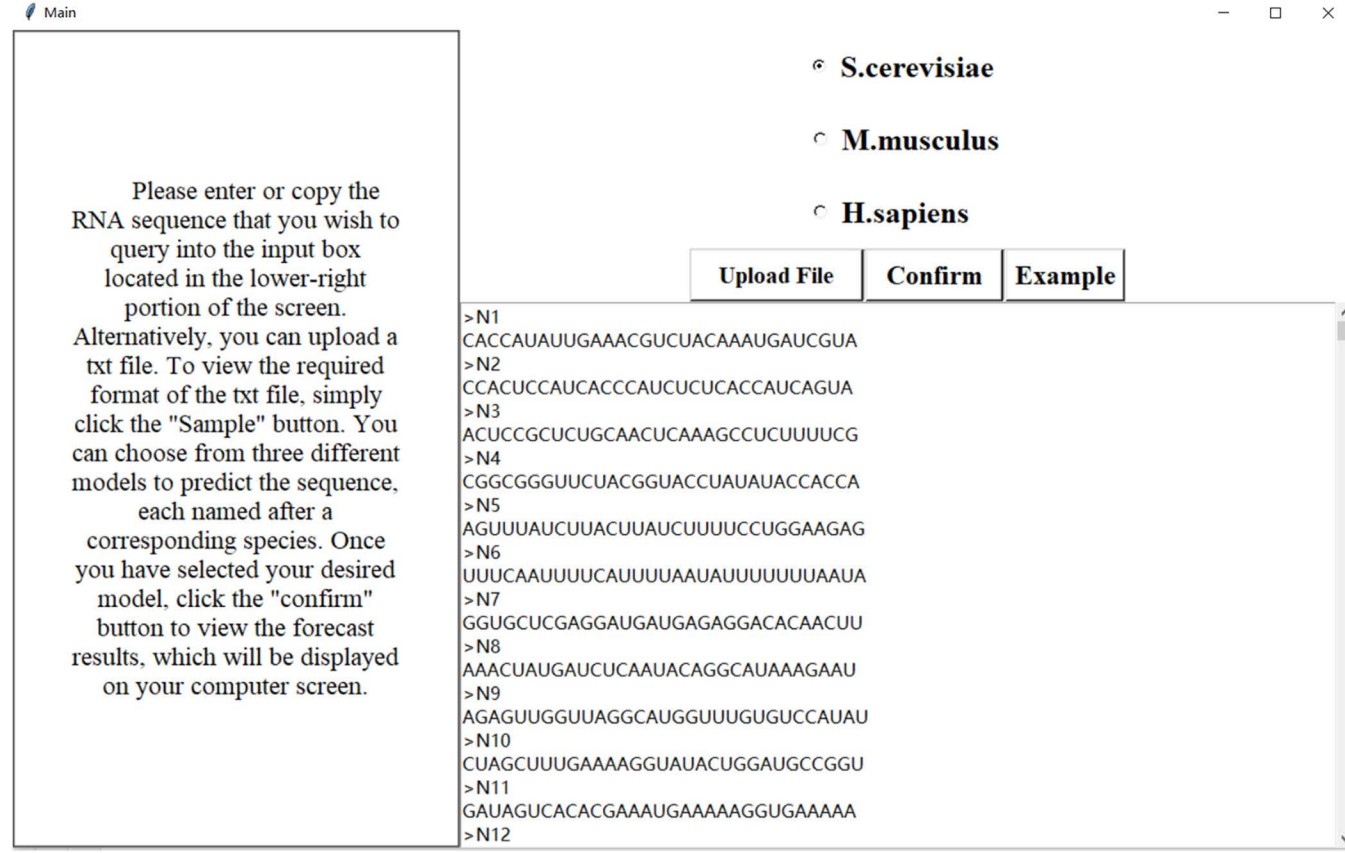

**Fig 4. Main interface of the software.**

in the existing bioinformatics tool landscape and offering a novel approach to this specific aspect of RNA sequence analysis.

On the main interface, users can enter or copy the RNA sequence they want to query into the input box in the lower right corner, or upload a txt file. Clicking the "example" button allows users to view the required format for the txt file. After a file is successfully uploaded, the data content will be displayed in the lower right box. The software provides three models for users to choose from, named after the corresponding species: *H. sapiens*, *S. cerevisiae*, and *M. musculus*. When users click the "Confire" button above the text box, the model will analyze and calculate the sequence, and return the sequence's name, length, and whether it contains a site to the user. After the prediction is complete, users can click the "download" button to export the prediction result file to a specified path. The prediction result interface is shown in Fig 5. Please note that there should be no grammar errors.

| Sequence | Number of nucleotides | Site |
|---|---|---|
| >H.cerevisiae_P1 | 31 | Yes |
| >H.cerevisiae_P2 | 31 | Yes |
| >H.cerevisiae_P3 | 31 | No |
| >H.cerevisiae_P4 | 31 | No |
| >H.cerevisiae_P5 | 31 | Yes |
| >H.cerevisiae_P6 | 31 | Yes |
| >H.cerevisiae_P7 | 31 | Yes |
| >H.cerevisiae_P8 | 31 | No |
| >H.cerevisiae_P9 | 31 | No |
| >H.cerevisiae_P10 | 31 | Yes |
| >H.cerevisiae_P11 | 31 | Yes |
| >H.cerevisiae_P12 | 31 | Yes |
| >H.cerevisiae_P13 | 31 | Yes |
| >H.cerevisiae_P14 | 31 | No |
| >H.cerevisiae_P15 | 31 | Yes |
| >H.cerevisiae_P16 | 31 | Yes |
| >H.cerevisiae_P17 | 31 | Yes |
| >H.cerevisiae_P18 | 31 | Yes |
| >H.cerevisiae_P19 | 31 | No |
| >H.cerevisiae_P20 | 31 | No |
| >H.cerevisiae_P21 | 31 | No |
| >H.cerevisiae_P22 | 31 | Yes |
| >H.cerevisiae_P23 | 31 | Yes |
| >H.cerevisiae_P24 | 31 | Yes |
| >H.cerevisiae_P25 | 31 | No |
| >H.cerevisiae_P26 | 31 | No |
| >H.cerevisiae_P27 | 31 | No |
| >H.cerevisiae_P28 | 31 | No |
| >H.cerevisiae_P29 | 31 | No |
| >H.cerevisiae_P30 | 31 | No |

**Download**

**Fig 5. Prediction results interface.**

## 4. Discussion

The accurate identification of pseudouridine sites is of great significance as it is involved in numerous crucial biological processes. In this study, we developed a novel computational method, Definer, to address the challenge of identifying RNA pseudouridine loci in H. sapiens, S. cerevisiae, and M. musculus.

With the explosive growth of genomics data, the limitations of traditional experimental methods for identifying RNA modification sites have become increasingly prominent. Computational methods have emerged as a powerful alternative. Our proposed Definer method combines two sequence coding schemes, NCP and One-hot, which allows for a more comprehensive representation of RNA sequence features. By feeding these features into a deep learning model composed of CNN, GRU, and Attention, we were able to capture both local and global sequence information, as well as the importance of different regions within the sequence.

The benchmark dataset, which includes data from three species, provided a solid foundation for evaluating the performance of Definer. The results of 10-fold cross-validation clearly demonstrated that Definer outperforms other existing methods. This superiority can be attributed to the effective combination of the sequence coding schemes and the powerful deep learning architecture. The independent testing of the data sets of H. sapiens and S. cerevisiae further validated the robustness and predictive ability of the model.

However, it is important to note that there are still some limitations and areas for improvement. For example, although Definer has shown good performance, the complexity of biological systems means that there may be other factors that affect pseudouridine site identification that have not been fully considered. Future studies could explore incorporating additional types of data, such as structural information or epigenetic marks, to further enhance the accuracy of the model.

In addition, as new experimental techniques for detecting pseudouridine sites are developed, the benchmark datasets may need to be updated and refined to ensure the continued relevance and effectiveness of computational methods like Definer. Overall, our study represents an important step forward in the accurate identification of RNA pseudouridine sites, and we believe that Definer has the potential to be a valuable tool in further understanding the functional mechanisms of this important modification site and its implications in various biological processes and diseases.

## 5. Conclusion

In this research, we have tackled the crucial task of identifying RNA pseudouridine sites. Pseudouridine, being widely distributed in non-coding RNAs and implicated in essential biological functions and diseases, demands accurate identification to decipher its functional mechanisms. The exponential growth of genomics data has necessitated the development of computational approaches, as traditional experimental methods have become insufficient.

Our proposed method, Definer, offers a promising solution. By integrating two sequence coding strategies, NCP and One-hot, it is capable of extracting comprehensive RNA sequence features. These features are then processed by a deep learning model composed of CNN, GRU, and Attention, which effectively capture the complex patterns and relationships within the RNA sequences.

The evaluation using a benchmark dataset covering three species, H. sapiens, S. cerevisiae, and M. musculus, and the application of 10-fold cross-validation have provided robust evidence that Definer outperforms existing methods. The independent testing of the datasets from H. sapiens and S. cerevisiae further bolsters the confidence in the model's predictive capabilities.

Overall, Definer represents a significant advancement in the field of RNA pseudouridine site identification. It has the potential to enhance our understanding of the role of pseudouridine in gene expression, RNA structural stability, and disease mechanisms. Future research could focus on further optimizing the method, exploring its application in other species or RNA types, and investigating potential synergies with other omics data to provide a more holistic view of the complex regulatory networks involving pseudouridine. With continued development and refinement, Definer could become an invaluable tool in both basic biological research and clinical applications related to RNA modifications.

## Author contributions

**Conceptualization:** Bo Han.

**Data curation:** Yang Liu.

**Formal analysis:** Jiezhang Wu.

**Investigation:** Sudan Bai.

**Methodology:** Bo Han, Sudan Bai.

**Software:** Yang Liu.

**Validation:** Yang Liu, Ruihao Xin.

**Visualization:** Jiezhang Wu.

**Writing – original draft:** Jiezhang Wu, Xin Feng.

**Writing – review & editing:** Bo Han, Ruihao Xin.

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
