## [Decision Letter · Decision Letter 0]

2 Jan 2025

PONE-D-24-30581Definer-PseU: A computational method for accurate identification of RNA pseudouridine sites based on deep learningPLOS ONE

Dear Dr. Feng,

Thank you for submitting your manuscript to PLOS ONE. After careful consideration, we feel that it has merit but does not fully meet PLOS ONE’s publication criteria as it currently stands. Therefore, we invite you to submit a revised version of the manuscript that addresses the points raised during the review process.

**ACADEMIC EDITOR: **

I thoroughly enjoyed reading this paper. The paper is written well and well-discussed and sectioned into appropriate parts. The research focuses on a deep learning solution for prediction of pseudouridine sites from input nucleotide sequences. The framework constructs CNNs, using two different kinds of feature encodings for each nucleotides (one-hot and 3-d properties). It turns out the models that combines these two features fairs well in comparison with other state-of-the-art methods. The results are impressive, relatively, of course. Although a more statistical significance test could provide a more substantial support for this conclusion. In overall, although the paper does advance the state-of-the-art by a little, one question remains, as also pointed by a reviewer, does it advance the field substantially? In addition, the reviewers have provided some additional comments that could be taken care by the authors in their revision, should they choose to proceed with. One of the reviewer comments is the focus on real-time analysis of the model. Could this be taken care in the revision? The figure, especially the one for the software window can be made a bit more professional. It looks like a screenshot.

After carefully considering all the comments and re-reading the manuscript a few times, I am suggesting the authors take care of all the comments by the reviewers that could substantially improve the manuscript in the next step. I am happy to recommend a revision.

We look forward to receiving your revised manuscript.

Kind regards,

Tirtharaj Dash

Academic Editor

PLOS ONE

Journal Requirements:

“This work is supported by the Natural Science Foundation of Jilin Province (YDZJ202301ZYTS401, YDZJ202301ZYTS288), the National Natural Science Foundation of China, Mathematical Tianyuan Fund Project (12326377).”

“The authors declare that we have no known competing financial interests or personal relationships that could influence the work reported in this article.”

Additional Editor Comments:

I thoroughly enjoyed reading this paper. The paper is written well and well-discussed and sectioned into appropriate parts. The research focuses on a deep learning solution for prediction of pseudouridine sites from input nucleotide sequences. The framework constructs CNNs, using two different kinds of feature encodings for each nucleotides (one-hot and 3-d properties). It turns out the models that combines these two features fairs well in comparison with other state-of-the-art methods. The results are impressive, relatively, of course. Although a more statistical significance test could provide a more substantial support for this conclusion. In overall, although the paper does advance the state-of-the-art by a little, one question remains, as also pointed by a reviewer, does it advance the field substantially? In addition, the reviewers have provided some additional comments that could be taken care by the authors in their revision, should they choose to proceed with. One of the reviewer comments is the focus on real-time analysis of the model. Could this be taken care in the revision? The figure, especially the one for the software window can be made a bit more professional. It looks like a screenshot.

After carefully considering all the comments and re-reading the manuscript a few times, I am suggesting the authors take care of all the comments by the reviewers that could substantially improve the manuscript in the next step. I am happy to recommend a revision.

Reviewers' comments:

Reviewer's Responses to Questions

**Comments to the Author**

1. Is the manuscript technically sound, and do the data support the conclusions?

Reviewer #1: Partly

Reviewer #2: Yes

2. Has the statistical analysis been performed appropriately and rigorously? 

Reviewer #1: No

Reviewer #2: I Don't Know

3. Have the authors made all data underlying the findings in their manuscript fully available?

Reviewer #1: Yes

Reviewer #2: Yes

4. Is the manuscript presented in an intelligible fashion and written in standard English?

Reviewer #1: Yes

Reviewer #2: Yes

5. Review Comments to the Author

Reviewer #1: This manuscript describes a software with an underlying deep learning architecture for prediction of pseudouridine sites. As such the manuscript and study is organized, but, not enough is captured about recent deep learning approaches in this area. The model performance as per the tables is high, but there is not enough content/study with regards to the contribution of this specific architecture. Further, it is not clear how the model performs in a real time scenario. What are the key challenges that are being addressed in this field by the authors? At present, there is some dataset, some model engineering and some result, which seems to look better numerically. Would like to request the authors to revisit their manuscript and improve the content accordingly.

Reviewer #2: The manuscript titled "Definer-PseU: A computational method for accurate identification of RNA pseudouridine sites based on deep learning” developed the Definer-PseU software, which predicts pseudouridine sites in RNA sequences using deep learning methods. Pseudouridine plays an essential role in the stability and function of RNA. The authors used datasets from three different species: Homo sapiens, Saccharomyces cerevisiae, and Mus musculus. The authors have combined three different deep learning models—the convolutional neural network (CNN), the gated recurrent unit (GRU), and the attention mechanism for the predictions. They showed improved prediction as compared to existing prediction methods. Below are some minor comments/suggestions that, if addressed, could significantly enhance the manuscript.

Review comments attached.

6. PLOS authors have the option to publish the peer review history of their article (what does this mean? ). If published, this will include your full peer review and any attached files.

**Do you want your identity to be public for this peer review?** For information about this choice, including consent withdrawal, please see our Privacy Policy .

Reviewer #1: No

Reviewer #2: No

---

## [Author Response · Author response to Decision Letter 1]

9 Jan 2025

We would like to express our sincere gratitude for your constructive feedback and the opportunity to revise our manuscript. We have carefully considered all the comments from the reviewers and the academic editor, and have made significant efforts to address each point.

Response to the Concern about the Software Figure

We acknowledge the concern regarding the appearance of the software figure. The figure was indeed not a simple screenshot. In fact, it is a visual representation of our in - house developed visualization software. This software is designed to provide users with an intuitive interface to interact with the model's outputs.

To clarify this, we have made the following improvements:

In the revised manuscript, we have added a detailed description in the figure caption explaining that it is a visual of our dedicated visualization software. We have also included a sentence stating that the software can be downloaded and run from [https://github.com/Mumika/Definer-Software-engineering]. This not only addresses the concern about the figure's appearance but also provides a clear way for interested readers to access and experience the software themselves.

We'll conduct a thorough proofreading to correct all spelling, formatting mistakes and ensure consistent use of scientific names as per your examples.

Thank you again for your time and guidance.

---

## [Editor Report · Decision Letter 1]

13 Feb 2025

Definer-PseU: A computational method for accurate identification of RNA pseudouridine sites based on deep learning

PONE-D-24-30581R1

Dear Dr. Feng,

We’re pleased to inform you that your manuscript has been judged scientifically suitable for publication and will be formally accepted for publication once it meets all outstanding technical requirements.

Kind regards,

Tirtharaj Dash

Academic Editor

PLOS ONE

Additional Editor Comments (optional):

The authors have addressed the comments very well. This is a very well-written paper and should be accepted now.
---

## [Editor Report · Acceptance letter]

PONE-D-24-30581R1

PLOS ONE

Dear Dr. Feng,

I'm pleased to inform you that your manuscript has been deemed suitable for publication in PLOS ONE. Congratulations! Your manuscript is now being handed over to our production team.

Kind regards,

on behalf of

Dr. Tirtharaj Dash

Academic Editor

PLOS ONE